# Cardiovascular risk in newly diagnosed type 2 diabetes patients in India

A. G. Unnikrishnan[1], R. K. Sahay[2], Uday Phadke[3], S. K. Sharma[4], Parag Shah[5], Rishi Shukla[6], Vijay Viswanathan[7], S. K. Wangnoo[8], Santosh Singhal[9], Mathew John[10], Ajay Kumar[11], Mala Dharmalingam[12], Subodh Jain[13], Shehla Shaikh[14], Willem J. Verberk[15]*

1 Chellaram Diabetes Institute, Pune, India, 2 Department of Endocrinology, Osmania Medical College, Osmania General Hospital, Hyderabad, India, 3 Sahyadri Hospital, Erandwane, Pune, Maharashtra, India, 4 Galaxy Speciality Centre, Jaipur, Rajasthan, India, 5 Gujarat Endocrine Centre, Ahmedabad, Gujarat, India, 6 Dr Rishi Shukla Centre for Diabetes & Endocrine Diseases, Kanpur, Uttar Pradesh, India, 7 MV Hospital for Diabetes, Chennai, Tamilnadu, India, 8 Indraprashth Apollo Hospital, New Delhi, India, 9 Director, Sparsh Health Care & CHS Apple Hospital, Gwalior, Madhya Pradesh, India, 10 Providence Endocrine and Diabetes Specialty Centre, Trivandrum, India, 11 Diabetes Care & Research Centre, Patna, Bihar, India, 12 Bangalore Endocrinology & Centre, Bangalore, Karnataka, India, 13 Diabetes Care Centre, Prayagraj, Uttar Pradesh, India, 14 KGN Diabetes & Endocrinology Centre, Nagpada, Mumbai, Maharashtra, India, 15 CARIM School for Cardiovascular Diseases, Maastricht University, Maastricht, the Netherlands

* Willem.verberk@microlife.ch

**Data Availability Statement:** The data will be made available upon reasonable request. For this the corresponding author can be contacted (willem.verberk@microlife.ch).

## Abstract

### Background

Type 2 diabetes mellitus (T2DM) worldwide continues to increase, in particular in India. Early T2DM diagnosis followed by appropriate management will result in more cardiovascular event free life years. However, knowledge of the cardiovascular profile of newly diagnosed T2DM patients is still limited. The aim of this study was to understand the extent of cardiovascular disease (CVD) risk of newly diagnosed T2DM patients in India.

### Methods

A cross sectional observational study was conducted to evaluate clinical laboratory and socio-demographic parameters of 5,080 newly diagnosed T2DM patients (48.3 ± 12.8 years of age; 36.7% female). In addition, we determined their cardiovascular risk according to the guidelines of the Lipid Association of India (LAI) and the criteria of the QRISK3 score.

### Results

Of the newly T2DM diagnosed patients in India 2,007(39.5%) were classified as "High risk" and 3,073 (60.5%) were classified as "Very high risk" based on LAI criteria. On average, patients had 1.7 ± 0.9 major atherosclerotic cardiovascular disease (ASCVD) risk factors. Low HDL-C value was the most frequent major risk (2,823; 55.6%) followed by high age (2,502; 49.3%), hypertension (2,141; 42.1%), smoking/tobacco use (1,078; 21.2%) and chronic kidney disease stage 3b or higher (568; 11.2%). In addition, 4,192 (82.5%) patients appeared to have at least one cholesterol abnormality and, if the latest LAI recommendations are applied, 96.5% (4,902) presented with lipid values above recommended targets.

**Funding:** This study was funded by Eris Lifesciences Ltd. Ahmedabad, India. The funders had no role in study design, data collection and analysis, decision to publish, or preparation of the manuscript.

**Competing interests:** The authors have declared that no competing interests exist.

Based on the QRISK3 calculation Indian diabetes patients had an average CVD risk of 15.3 ± 12.3%, (12.2 ± 10.1 vs. 17.1 ± 13.5 [*p*<0.001] for females and males, respectively).

## Conclusions

Newly diagnosed Indian T2DM patients are at high ASCVD risk. Our data therefore support the notion that further extension of nationwide ASCVD risk identification programs and prevention strategies to reduce the occurrence of cardiovascular diseases are warranted.

## Introduction

The prevalence of type 2 diabetes mellitus (T2DM) worldwide continues to increase, in particular in India where the prevalence of adults (>18 years) with diabetes increased in both rural and urban India from 2.4% and 3.3% in 1972 to 15.0% and 19.0%, respectively in the year 2015–2019 [1, 2]. This means an increase from approximately 6.5 to 68 million in rural and from approximately 2.3 to 9 million urban adults, thus presently a total of 107 million Indian adults have been diagnosed with T2DM. The prevalence of adults with pre-diabetes is much higher and is estimated to be 255 million (rural: 183 million and urban: 72 million) [1].

These alarming figures cause a burden to developing economies as the disease mainly affects those individuals who are most productive to support economic growth. In addition, rates of atherosclerotic cardiovascular disease (ASCVD) are strikingly high in India compared to Western countries [3]. For example, global burden figures from 2019 suggest that the prevalence of age standardized years of life lost due to cardiovascular diseases is at least two times higher in South Asia as compared to Western Europe and Australia [4]. T2DM largely contributes to this burden with more than 70% of patients with T2DM dying due to ASCVD [5].

DM in general confers to a two-fold excess risk of vascular outcomes (coronary artery disease, ischaemic stroke, and vascular deaths) independent of other risk factors, with a relative higher risk for women than men and with early onset of diabetes [6].

T2DM combined with target organ damage or with three or more major risk factors cause very high risk; a 10-year risk of cardiovascular disease (CVD) death higher than 10%, whereas most western T2DM patients are categorized as high risk (10-year risk of CVD death between 5 and 10%) [7].

An earlier diagnosis of T2DM followed by appropriate management of diabetes will result in more cardiovascular event free life years [8]. For this knowledge of the cardiovascular profile of patients with T2DM is needed together with a risk prediction model to estimate a patient's CVD risk. However, most of these models are based on populations from high-income countries, whereas there are differences in CVD risk depending on regions and ethnicities [9].

To reduce the CVD risk of T2DM patients, multifactorial intervention is needed. This approach includes lipid management, glycaemic control, control of hypertension, smoking cessation, weight reduction and increased physical activity. Optimum lipid lowering therapy treatment leads to a significant CVD risk reduction. Therefore, the Lipid Association of India (LAI) has recently proposed more strict and better structured treatment goals based on the Indian patients' individual CVD risk [10, 11]. The present study aims at investigating the CVD risk and recommended management of dyslipidaemia following these LAI recommendations in a population newly diagnosed T2DM patients in India. In addition, CVD risk was also calculated according to QRISK3, in order to determine the CVD risk related to individual risk factors and different ethnicities [12].

## Materials and methods

Patients with newly (i.e. first time) diagnosed T2DM identified in the period between 2017 and 2020 were analysed with the aim to learn more about their phenotypic characteristics in the Indian population. Patients were recruited in 928 medical centres by 1932 physicians (S1 File) based in 121 cities throughout 27 Indian states.

All patients gave written and signed informed consent and the protocol was approved by the independent Astha medical ethics committee, Ahmedabad, India (S2 File) and from Chellaram Diabetes Institute, Pune, India (S3 File).

All 5,080 patients underwent blood pressure measurements (two measurements in sitting position with one-minute interval time), weight and height measurements according to routine clinical practice and were interviewed on lifestyle (smoking, tobacco use, alcohol etc.), medical treatments and family related diseases. Thereafter, blood samples were drawn for lipids: total cholesterol (Cholesterol Oxidase, Esterase, Peroxide) high density lipoprotein (HDL, [Direct Measure, PEG]), low density lipoprotein (LDL [Direct Measure]) and triglycerides (TG [Enzymatic End Point]). In addition, serum creatinine (Alkaline Picrate Kinetic, IFC-C-IDMS Standardized) and glycemia related data were determined. The protocol and case record forms of the present study is shown in S2 File. All blood samples were analysed in a central certified lab.

### Statistical analysis

**Risk factor determination and risk classification.** For the present paper CVD risk was also calculated according to the QRISK3 chart [12] because it is the most suitable for Indians, as considered by the LAI [11]. The QRISK3 algorithm calculates a person's risk of developing a heart attack or stroke over the next 10 years. The advantage of QRISK3 as compared to other prediction models is that it considers different ethnicities, including the Indian so that the overall cardiovascular risk of the Indian population, specifically, can be estimated. Another advantage of QRISK3 is that the impact of individual major ASCVD risk factors on total risk can be estimated.

Lipid abnormalities were determined according to recommendations of LAI [10, 11]. The following cut-offs were used to identify abnormal levels: HDL < 40 mg/dL for males HDL<50 mg/dL for females, LDL ≥130 mg/dL, total cholesterol ≥200 mg/dL, TG ≥150 m/dL. Non-HDL was calculated as total cholesterol—HDL.

Hypertension was defined as a blood pressure higher than 140/90 mmHg or if the patient received anti-hypertensive treatment. Classification of chronic kidney disease was based on estimated glomerular filtration rated (eGFR). eGFR was calculated using the Chronic Kidney Disease Epidemiology Collaboration (CKD-EPI) equation using measured serum creatinine (Scr) and using the formula.: 175 × (SCr)-1.154 × (age)-0.203 × (0.742 if female) × (1.212 if Black) [13].

Patient's ASCVD risk classification was determined based on available parameters using the criteria of the LAI [10, 11]. Patients were classified as "high risk" (HR) and "very high risk" (VHR). According to LAI criteria a patient with DM is classified as HR or worse. VHR was defined as patients with DM and two more major ASCVD risk factors. The major ASCVD risk factors as defined by LAI and which were used in the present analysis are: age ≥45 years in males and ≥55 years in females, hypertension, smoking or tobacco use and low HDL-C (<40 mg/dL in males and <50 mg/dL in females). For the present study, renal impairment was the only measure of target organ damage that could be determined. Patients with T2DM and the presence of target organ damage (CKD 3b or higher defined as eGFR <45 mL/min/1.73 m2) were also classified as VHR.

Main demographic and clinical data were summarised by calculating the mean (±SD) in case of continuous variables and the absolute (n) and relative (%) frequency in case of categorical variables. Differences across groups were evaluated using analysis of variance (ANOVA) for continuous variables. For categorical variables, differences across groups were evaluated using Chi-square tests.

For the few missing values, population average values were imputed, there were no categorical values missing. QRISK3 allows only certain ranges, for example, age ranges from 25 to 80 years, lower values were changed for the lowest allowable value and higher values for the highest allowable age. This was restricted to QRISK3 calculations only.

Results were presented in $p$-values if considered relevant, a $p$-value of $<0.05$ was considered significant. Analyses were performed using the statistical package RStudio Version 1.2.5033 for Windows, for calculating QRISK-score, the QRISK3 package was used.

## Results and discussion

### Results

In total, 5,080 newly diagnosed patients were analysed with an average (± SD) age of 48.3 ± 12.8 years, an average HBA1c value of 8.1% and 36.7% (n = 1,864) were women (Table 1). Men had more major ASCVD risk factors than women (1.8 ± 1.0 vs. 1.5± 0.8) and consequently were more often classified as 'very high risk' (VHR) than women (63.1 vs. 56.0%). The higher risk of males as compared to females was due the fact that, for males, age is considered a major risk factor from the age of 45 years, whereas for females this is from the age of 55 years, males more often smoked or used tobacco (28.6 vs. 8.4%) and more often had hypertension (44.9% vs. 37.4%). Because the recommended threshold value for low HDL is higher for females than for males, females more often had HDL values below target value than males (76.9 vs. 43.2%). Females more often had an eGFR value below 45 mL/min/1.73 m$^2$ than males (CKD stage 3b or higher [14.9 vs. 9.1%], all $p<0.001$).

### Lipid profile

Table 2 shows that 4,192 (82.5%) patients had at least one lipid abnormality, with more patients being affected in the VHR group than in the 'high risk' (HR) group: 2,699 (87.8%) vs. 1,493 (74.4%, $p <0.001$).

The most prevalent abnormal cholesterol level was low HDL (55.6%), followed by high TG (51.3%), high total cholesterol (27.5%) and high LDL (26.8%).

In the HR group 1,839 (91.6%) patients had lipid values higher than the LAI recommended treatment target (non-HDL≥100 and LDL≥70 mg/dl), in the VHR group 3,056 (99.4%) had lipid levels above non-HDL≥80 and LDL≥50, which are the recommended treatment targets for this group.

Of all patients who met the group-specified recommended targets, 6 patients in the HR-group and 1 patient in the VHR-group, were on lipid lowering therapy. This means that, altogether, 1,845 (91.9%) patients in the HR-group, 3,057 (99.5%) in the VHR-group and 4,902 (96.5%) patients in total, had lipid values above threshold values as recommended by the LAI.

### QRISK3

Based on the QRISK3 risk chart the newly diagnosed Indian T2DM patients had an average QRISK3-score of 15.3 ±12.6% with females having a lower risk than males (12.2 ±10.1% vs. 17.1 ±13.5%, $p <0.001$). In addition, Table 3 shows that smokers had the highest CV risk, followed by CKD stage 3–5, hypertension, obesity and male gender.

**Table 1. Atherosclerotic risk factors separated according to sex, qualitative variables.**

| | Female (N = 1864) | Male (N = 3216) | Total (N = 5080) | p value |
|---|---|---|---|---|
| **Age [years]** | | | | 0.662 |
| | 48.2 (12.8) | 48.4 (12.9) | 48.3 (12.8) | |
| **BMI [kg/m2]** | | | | 0.939 |
| | 27.2 (4.8) | 27.2 (4.4) | 27.2 (4.6) | |
| **HbA1c [%]** | | | | 0.235 |
| | 8.1 (2.2) | 8.2 (2.2) | 8.1 (2.2) | |
| **Risk groups (LAI)*** | | | | < 0.001 |
| High risk | 820 (44.0%) | 1187 (36.9%) | 2007 (39.5%) | |
| Very high risk | 1044 (56.0%) | 2029 (63.1%) | 3073 (60.5%) | |
| **cholesterol abnormality ≥ 1** | | | | < 0.001 |
| No | 186 (10.0%) | 702 (21.8%) | 888 (17.5%) | |
| Yes | 1678 (90.0%) | 2514 (78.2%) | 4192 (82.5%) | |
| **Major ASCVD risk factors** | | | | < 0.001 |
| | 1.5 (0.8) | 1.8 (1.0) | 1.7 (0.9) | |
| **Age [y] as risk factor** | | | | < 0.001 |
| male <45, female <55 | 1285 (68.9%) | 1293 (40.2%) | 2578 (50.7%) | |
| male ≥45, female ≥55 | 579 (31.1%) | 1923 (59.8%) | 2502 (49.3%) | |
| **HDL [mg/dL]** | | | | < 0.001 |
| Normal | 430 (23.1%) | 1827 (56.8%) | 2257 (44.4%) | |
| Low (males <40, females <50) | 1434 (76.9%) | 1389 (43.2%) | 2823 (55.6%) | |
| **Smoking** | | | | < 0.001 |
| No | 1820 (97.6%) | 2565 (79.8%) | 4385 (86.3%) | |
| Yes | 36 (1.9%) | 532 (16.5%) | 568 (11.2%) | |
| Ex-smoker | 8 (0.4%) | 119 (3.7%) | 127 (2.5%) | |
| **Tobacco** | | | | < 0.001 |
| No | 1743 (93.5%) | 2319 (72.1%) | 4062 (80.0%) | |
| Yes | 121 (6.5%) | 897 (27.9%) | 1018 (20.0%) | |
| **Smoking/tobacco use** | | | | < 0.001 |
| No | 1707 (91.6%) | 2295 (71.4%) | 4002 (78.8%) | |
| Yes | 157 (8.4%) | 921 (28.6%) | 1078 (21.2%) | |
| **Hypertension (Total)** | | | | < 0.001 |
| No | 1167 (62.6%) | 1772 (55.1%) | 2939 (57.9%) | |
| Yes | 697 (37.4%) | 1444 (44.9%) | 2141 (42.1%) | |
| **eGFR [mL/min/1.73 m$^2$]** | | | | < 0.001 |
| <45 | 275 (14.8%) | 293 (9.1%) | 568 (11.2%) | |
| ≥45 | 1589 (85.2%) | 2923 (90.9%) | 4512 (88.8%) | |
| **Cholesterol lowering treatment** | | | | 0.010 |
| No | 1748 (93.8%) | 2952 (91.8%) | 4700 (92.5%) | |
| Yes | 116 (6.2%) | 264 (8.2%) | 380 (7.5%) | |
| **Anti-Hypertension treatment** | | | | < 0.001 |
| No | 1606 (86.2%) | 2636 (82.0%) | 4242 (83.5%) | |
| Yes | 258 (13.8%) | 580 (18.0%) | 838 (16.5%) | |

BMI indicates body mass index;

*Risk group classification based on the recommendations of the Lipid association of India;

eGFR, estimated glomerular filtration rate.

**Table 2. Lipid parameters separated for LAI risk categories.**

| | High risk (N = 2007) | Very high risk (N = 3073) | Total (N = 5080) | p value |
|---|---|---|---|---|
| **Cholesterol abnormality ≥ 1** * | | | | < 0.001 |
| No | 514 (25.6%) | 374 (12.2%) | 888 (17.5%) | |
| Yes | 1493 (74.4%) | 2699 (87.8%) | 4192 (82.5%) | |
| **HDL [mg/dL]** | | | | < 0.001 |
| Normal | 1280 (63.8%) | 977 (31.8%) | 2257 (44.4%) | |
| Low (males <40, females <50) | 727 (36.2%) | 2096 (68.2%) | 2823 (55.6%) | |
| **LDL [mg/dL]** | | | | 0.302 |
| <50 | 28 (1.4%) | 32 (1.0%) | 60 (1.2%) | |
| 50–70 | 178 (8.9%) | 251 (8.2%) | 429 (8.4%) | |
| 70–100 | 603 (30.0%) | 999 (32.5%) | 1602 (31.5%) | |
| 100–130 | 648 (32.3%) | 980 (31.9%) | 1628 (32.0%) | |
| ≥130 | 550 (27.4%) | 811 (26.4%) | 1361 (26.8%) | |
| **non-HDL [mg/dL]** | | | | 0.050 |
| ≥100 | 1428 (71.2%) | 2282 (74.3%) | 3710 (73.0%) | |
| 80–100 | 268 (13.4%) | 371 (12.1%) | 639 (12.6%) | |
| <80 | 311 (15.5%) | 420 (13.7%) | 731 (14.4%) | |
| **Total Cholesterol [mg/dL]** | | | | 0.019 |
| normal[<200] | 1419 (70.7%) | 2265 (73.7%) | 3684 (72.5%) | |
| dyslipidemia[≥200] | 588 (29.3%) | 808 (26.3%) | 1396 (27.5%) | |
| **Triglycerides [mg/dL]** | | | | < 0.001 |
| normal [<150] | 1052 (52.4%) | 1424 (46.3%) | 2476 (48.7%) | |
| borderline high [150–200] | 513 (25.6%) | 823 (26.8%) | 1336 (26.3%) | |
| high[200–500] | 413 (20.6%) | 768 (25.0%) | 1181 (23.2%) | |
| very high[≥500] | 29 (1.4%) | 58 (1.9%) | 87 (1.7%) | |
| **LAI treatment goals (mg/dL)** | | | | |
| non-HDL<100 & LDL<70 | 168 (8.4%) | - | - | |
| non-HDL≥100 or LDL≥70 | 1839 (91.6%) | - | - | |
| **LAI treatment goals (mg/dL)** | | | | |
| non-HDL<80 & LDL<50 | - | 17 (0.6%) | - | |
| non-HDL≥80 or LDL≥50 | - | 3056 (99.4%) | - | |

LAI indicates Lipids association of India; HDL, high density lipoproteins; LDL, low density lipoproteins.

* at least one of the lipid parameters (total cholesterol, LDL, HDL or Triglycerides) above normal value as recommended by LAI.

Fig 1 shows that the cardiovascular risk strongly increases with age and after 35 years of age males have significantly higher risk than males.

## Discussion

Newly diagnosed T2DM patients overall had a high risk with 40% of all patients being classified as HR and 60% classified as VHR. Males were more often classified as VHR than females. The most common CV risk factor was a low HDL value according to LAI criteria with, 68% of all subjects appeared to have at least one lipid abnormality. Based on the strict criteria of the LAI all newly diagnosed T2DM patients require cholesterol lowering treatment. In the present population, almost all (97% of all) newly diagnosed T2DM Indian patients had cholesterol levels above the recommended threshold values. According to the QRISK3-score smokers had 7% higher CV risk than non-smokers and hypertensives almost 5% higher risk than normotensives.

**Table 3. Qrisk3-score (%) per group separated for sex and major atherosclerotic cardiovascular risk factors, Mean (SD).**

| Females (N = 1864) | Males (N = 3216) | p value |
|---|---|---|
| 12.2 (10.1) | 17.1 (13.5) | < 0.001 |
| Non-smokers (N = 4512) | Smokers (N = 568) | |
| 14.5 (12.2) | 21.4 (13.8) | < 0.001 |
| Normotension (N = 3567) | Hypertension* (N = 1513) | |
| 13.9 (12.1) | 18.5 (13.0) | < 0.001 |
| Normal (N = 4056) | Obese (N = 1024) | |
| 14.8 (12.5) | 17.1 (12.8) | < 0.001 |
| CKD [stage 1,2] (N = 3951) | CKD [stage 3–5] (N = 1129) | |
| 13.7 (11.8) | 20.8 (13.6) | < 0.001 |

*Hypertension based on systolic blood pressure and/or anti-hypertension treatment.

CKD, indicates chronic kidney disease.

Recently, analysis of a large Mediterranean T2DM patient database that classified patients according to the ESC chart [7], showed that most patients with T2DM (53.4%; 95% CI, 53.1% -53.6%) were at very high risk of fatal CV events and males were at higher risk than females (55.6% vs. 50.7%) [14]. The prevalence of patients with VHR was higher in the present Indian T2DM population than in the Mediterranean T2DM population despite the fact that the Mediterranean population was, on average, more than 20 years older than the Indian population (70.1± 12.3 vs. 48.1 ± 12.8) and despite the fact that the entire population of T2DM patients was considered instead of only the newly diagnosed ones as was done in the present study.

The reason for this high risk in the present population is related to the stricter classification of LAI as compared to the ESC classification, due to the higher CV risk of Indians as compared to the Caucasian population. For example, as Indians get coronary artery disease one or two

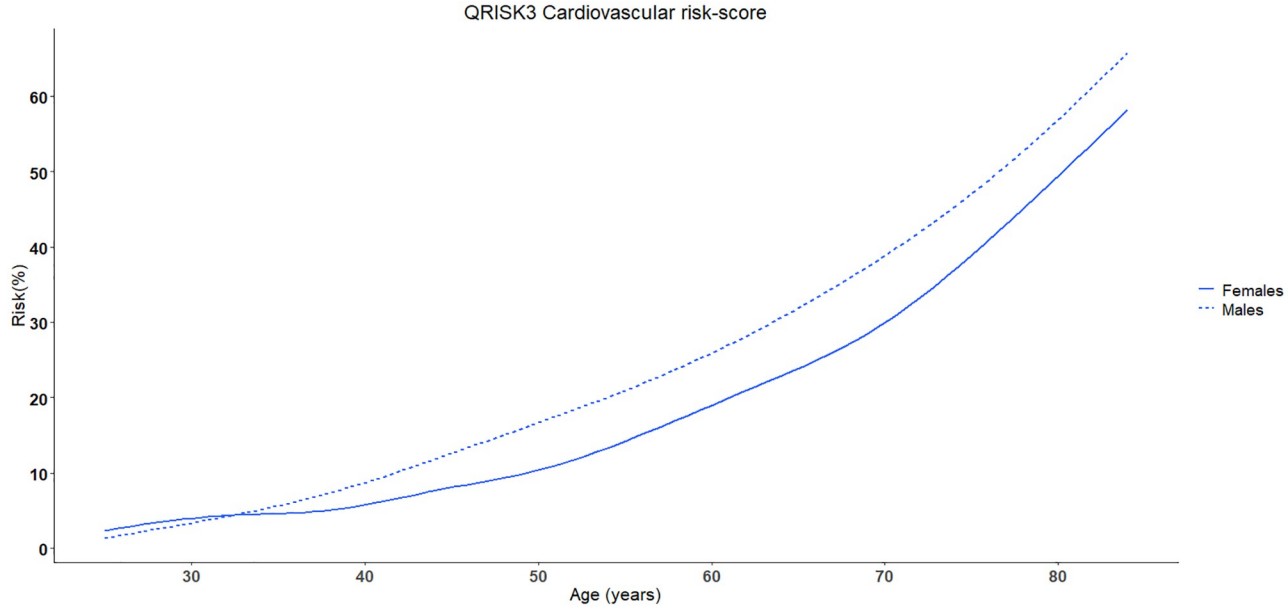

**Fig 1. Cardiovascular risk based on QRISK3 chart related to age and separated for sex.**

decades earlier than their western counterparts [15], being Indian male and older than 45 years of age is already considered a major risk factor according to LAI whereas according to ESC criteria this is a decade later. Also, the QRISK3-score ascribes more risk to the Indian population as compared to whites (3% higher) and black Africans (7% higher, see S1 Fig).

In addition, where the ESC mainly considers LDL as a major ASCVD risk factor, LAI mainly considers low HDL values as a major risk factor, especially for females [10]. Of the current population, 56% had lower HDL values than the recommended LAI threshold.

## Dyslipidaemia

The LAI recommends low lipid targets for HR (non-HDL<100 mg/dL and LDL<70 mg/dL) and VHR patients (non-HDL < 80 mg/dL or LDL< 50 mg/dL). T2DM patients are classified as HR or worse, which means that cholesterol lowering treatment is recommended. The recommendation for aggressive lipids lowering treatment is based on the evidence from a meta-analysis including 18,686 DM patients which showed that a statin-induced reduction of LDL by 40 mg/dL was associated with a 9% reduction in all-cause mortality and a 21% reduction in the incidence of major CV events [16].

## Hypertension

Of all patients, 42% (n = 4,049) presented with hypertension. This seems relatively low as the prevalence of hypertension in T2DM patients has been reported as high as 60% [17]. However, the present population is relatively young (<50 years), first time diagnosed with T2DM and most of them were not under physician's treatment before. In addition, blood pressure measurement was performed during the clinic visit whereas the diagnosis of hypertension should be based ideally on ambulatory blood pressure measurement, moreover because T2DM patients often are non-dippers and or have elevated night-time blood pressure [18, 19]. The current ESC guidelines recommend to lower systolic blood pressure below 130 mmHg, if well tolerated, for patients younger than the age of 65 years. In the present analysis 3,617 (37.3%) patients were hypertensive and younger than 65 years of age, which indicates that more intensified treatment is needed for this group.

## Obesity

According to the LAI consensus report obesity is considered a risk factor that deserves lifestyle modification, but it is not considered a major risk factor. According to the QRSIK3 calculation the obese patients in the current population had 17.1% risk as compared to 14.8% for those with lower BMI. However, as BMI has some limitations as a risk predictor it may be questioned if it is appropriate to determine CVD risk [20] based on BMI values for the current study population. Indians have a body composition that differs from other ethnicities [21] so that BMI may not be the best method to estimate CVD risk in Indians and / or that the threshold values for overweight and obesity may need to be lowered. In this case, using body fat measurement or waist to hip ratio may be preferred over BMI to estimate CVD risk.

## Smoking tobacco use

The prevalence of smoking in the present male population was high, which is not only a major ASCVD risk factor but may also have played a role in the onset of diabetes [22]. The QRISK3 chart considers smoking as an important risk factor but the LAI statement also considers the use of smokeless tobacco. In contradiction to most western countries use of smokeless tobacco is very common in India and more common than smoking in the present population (20% vs.

11%). Although, smokeless tobacco is marketed as a safe alternative for smoking in India [23] it can contain a much higher amount of nicotine than cigarettes (50 mg/g vs. 13 mg/rod for the highest nicotine content) [24] and therefore may significantly contribute to insulin resistance [25–27]. This asks for interventions to educate tobacco consumers about the harmful effects of using smokeless tobacco [23].

## Strength and limitations

To the best of our knowledge this is the first study of this size investigating the CVD risk factors of newly diagnosed patients with T2DM in India. Many biomarkers were measured so that a good overall CVD risk prediction could be established. However, this study should also be seen within the context of its limitations: Newly diagnosed T2DM was defined as the first time the patient was diagnosed with T2DM. However, some patients might have had T2DM for a long time already that was only recently discovered, whereas other patients were diagnosed relatively earlier. This might have caused some heterogeneity regarding cardiovascular risk profile within the present patient population. As is commonly seen in India, there were more males than females participating in the present study (64 vs. 36%). Because women in India have the tendency to seek healthcare support later than men, women possibly have a relatively higher risk than men in the present patient population. However, the numbers are such that valid conclusions can be drawn from the female data subset. As already mentioned, hypertension, an important CVD risk predictor, was largely based on blood pressure measurement that was taken in the clinic. This is liable to some degree of uncertainty and may have influenced risk calculation. On the other hand, this represents general clinical practice and is how clinic BP measurement is performed in India and it may possibly reveal the limitations of this standard procedure. Patients with T2DM and target organ damage (TOD) are categorized as VHR. However, in the present study several parameters were not considered. This might have caused underestimation of the number of VHR patients in the present analysis. India is a diverse country in terms of ethnicity, and it cannot be excluded that this might have some effect on cardiovascular risk. In addition, a recently published results of more than 100,000 Indians screened for CVD risk at airport locations showed that there were differences in BMI, blood glucose, and blood pressure values between some states of India [28]. Although, both ethnicity and region might be relevant and worth considering when verifying CVD risk among Indians, this was not done in the present study.

Finally, although the QRISK3 chart is considered by the LAI to be the most suitable for Indians [11], it has not been validated in Indians.

## Conclusion

This study describes the patient characteristics and associated CVD risk of patients newly diagnosed with T2DM in India. The presented data show that newly diagnosed T2DM is associated with, on average 1.7 major risk factors and that two-thirds of all patients are classified as VHR (60.5%) and the remainder as HR (39.5%).

Careful consideration of the cardiovascular risk burden of newly diagnosed T2DM patients is important as it will most likely lead to more intensified and more effective treatment leading to reduced risk of cardiovascular events [29]. Indians are uniquely vulnerable to cardiovascular disease, and the disease occurs almost a decade earlier in them [30]. In addition, it was shown that there is an elevated risk for cardiovascular disease, prior to T2DM diagnosis [31]. These results, taken together with results of our study suggest that appropriate and intensive management of cardiovascular risk factors is important in young people at risk of diabetes as well as in young people recently diagnosed with T2DM.

T2DM in India is a serious health issue. The attention for nationwide preventive and treatment strategies needs to be extended and/or implemented. This requires, next to glucose lowering treatment, aggressive lipid lowering treatment for all patients and approximately half of them would also need anti-hypertensive treatment. As the recommendation of (aggressive) lipid lowering treatment for all T2DM patients are relatively new, education is needed to improve awareness among both patients and physicians. Because CVD risk seems to differ among ethnicities a CDV risk chart, specific for the Indian population, is needed to properly estimate the individual patient CVD risk and improve targeted treatment strategy.

Our data support the notion that further extension of nationwide ASCVD risk identification programs and prevention strategies to reduce the occurrence of cardiovascular diseases are warranted. For this, long-term follow-up studies are needed that investigate the relationships between patient characteristics of newly diagnosed T2DM Indian patients in relation to CVD risk outcomes, preventive measures and treatment strategies.

## Supporting information

**S1 Fig. Average QRISK3 per ethnicity, risk score was calculated based on the parameters of the present population, with only "ethnicity" changed.** The stripes in the box plots represent median score, the dots represent average values as presented.
(PDF)

**S1 File. Names of participating doctors.**
(DOCX)

**S2 File. Study protocol India diabetes study.**
(PDF)

**S3 File. IDS study data.**
(CSV)

## Author Contributions

**Conceptualization:** Willem J. Verberk.

**Data curation:** A. G. Unnikrishnan, R. K. Sahay, Uday Phadke, S. K. Sharma, Parag Shah, Rishi Shukla, Vijay Viswanathan, S. K. Wangnoo, Santosh Singhal, Mathew John, Ajay Kumar, Mala Dharmalingam, Subodh Jain, Shehla Shaikh.

**Formal analysis:** Willem J. Verberk.

**Supervision:** A. G. Unnikrishnan.

**Writing – original draft:** Willem J. Verberk.

**Writing – review & editing:** A. G. Unnikrishnan, R. K. Sahay, Uday Phadke, S. K. Sharma, Parag Shah, Rishi Shukla, Vijay Viswanathan, S. K. Wangnoo, Santosh Singhal, Mathew John, Ajay Kumar, Mala Dharmalingam, Subodh Jain, Shehla Shaikh.

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
