## [Decision Letter · Decision Letter 0]

19 Dec 2021

PONE-D-21-35344Cardiovascular risk in newly diagnosed type 2 diabetes patients in India.PLOS ONE

Dear Dr. Verberk,

Thank you for submitting your manuscript to PLOS ONE. After careful consideration, we feel that it has merit but does not fully meet PLOS ONE’s publication criteria as it currently stands. Therefore, we invite you to submit a revised version of the manuscript that addresses the points raised during the review process.

We look forward to receiving your revised manuscript.

Kind regards,

Kanhaiya Singh, Ph.D

Academic Editor

PLOS ONE

Journal Requirements:

Additional Editor Comments:

Please address the concern raised by Reviewer 1 and Reviewer 2. Also please discuss more in detail about the significance of this study in the field of cardiovascular risk in newly diagnosed T2DM patients.

Reviewers' comments:

Reviewer's Responses to Questions

**Comments to the Author**

1. Is the manuscript technically sound, and do the data support the conclusions?

Reviewer #1: Yes

Reviewer #2: Yes

Reviewer #3: Yes

2. Has the statistical analysis been performed appropriately and rigorously? 

Reviewer #1: Yes

Reviewer #2: Yes

Reviewer #3: Yes

3. Have the authors made all data underlying the findings in their manuscript fully available?

Reviewer #1: Yes

Reviewer #2: No

Reviewer #3: Yes

4. Is the manuscript presented in an intelligible fashion and written in standard English?

Reviewer #1: Yes

Reviewer #2: Yes

Reviewer #3: Yes

5. Review Comments to the Author

Reviewer #1: In this study the authors are evaluating the risk of occurrence of cardiovascular diseases (CVD) in patients who have been recently diagnosed with Type 2 Diabetes Mellitus (T2DM). The evaluation shows that there is a high risk of CVD in patients with T2DM. The authors propose that this study should help raise awareness and increase risk identification strategies to help curb CVD in T2DM patients.

Overall, the manuscript is well written with a few grammatical errors (especially with the article ‘the’). However, there are a few suggestions. Please see a couple of points below:

Introduction, paragraph 2, what is the basis of the claim in line 1? Is there a missing citation?

Research design and methods, statistical analysis, the authors mention 10 year, whereas the study was conducted for patients diagnosed between 2017-2020. Please clarify.

What was the selection criteria for ‘newly diagnosed’? Were these patients diabetic for a long time and just discovered later? What were the Hb1Ac levels for these patients?

India is a very diverse country in terms of ethnicity. What were the ethnicities of the patients? Does that play a role in assessing the risk factors?

Page 15, last line mentions male twice instead of male and female.

Some subheadings are italicized while others are not. Please correct it.

Reviewer #2: Ref: PONE-D-21-35344

In the present article entitled “Cardiovascular risk in newly diagnosed type 2 diabetes patients in India” Unnikrishnan et al. have looked into the extent of cardiovascular disease (CVD) risk of newly diagnosed T2DM patients in India. The study design is simple and the results are well supported by the data.

The multicentric nature of the study makes the sample representative of the diverse population in a vast country like India. The limitations of the study have been laid out well.

Here are a few suggestions to make the manuscript more robust prior to publication:

1. Details of Funders of the study have not been included in the financial disclosures: 'Enter a financial disclosure statement that describes the sources of funding for the work included in this submission.'

2. Detailed ethics statement should be made available prior to final publishing since the study involves health data from multiple participants.

3. Kindly inform the editorial team how we can help with regards to making the underlying the results presented in the study available : '... but some help from the journal may be needed.'

4. Kindly submit supplementary data regarding patient details, blood tests done/other parameters calculated.

5. The names of the participating doctors/ scientists may be included in a supplementary document

Reviewer #3: In this manuscript authors are showing the cardiovascular risk assessment in newly diagnosed T2DM patients in India. Sample size and the guidelines to assess the risk is appropriate for this cross-sectional study. except minor edits, manuscript looks good.

6. PLOS authors have the option to publish the peer review history of their article (what does this mean?). If published, this will include your full peer review and any attached files.

Reviewer #1: No

Reviewer #2: No

Reviewer #3: No

---

## [Author Response · Author response to Decision Letter 0]

4 Jan 2022

Response to the reviewers

We highly thank the editor and reviewers for spending their valuable time to review our manuscript and for providing their comments, which helped to improve the quality of the manuscript. 

Below you can find how we responded to each point raised by the editor and reviewers. 

Additional Editor Comments:

Please address the concern raised by Reviewer 1 and Reviewer 2. 

Also please discuss more in detail about the significance of this study in the field of cardiovascular risk in newly diagnosed T2DM patients.

We thank the editor for the above comment and have added the following paragraph to the discussion section 

“Careful consideration of the cardiovascular risk burden of newly diagnosed T2DM patients will most likely lead to more intensified and more effective treatment leading to reduced risk of cardiovascular events [28]. Indians are uniquely vulnerable to cardiovascular disease, and the disease occurs almost a decade earlier in them [29]. In addition, it was shown that there is an elevated risk for cardiovascular disease, prior to T2DM diagnosis [30]. These results, taken together with results of our study suggest that appropriate and intensive management of cardiovascular risk factors is important in young people at risk of diabetes as well as in young people recently diagnosed with T2DM.”

Reviewer #1: In this study the authors are evaluating the risk of occurrence of cardiovascular diseases (CVD) in patients who have been recently diagnosed with Type 2 Diabetes Mellitus (T2DM). The evaluation shows that there is a high risk of CVD in patients with T2DM. The authors propose that this study should help raise awareness and increase risk identification strategies to help curb CVD in T2DM patients.

Overall, the manuscript is well written with a few grammatical errors (especially with the article ‘the’). However, there are a few suggestions. Please see a couple of points below:

We kindly thank the reviewer for the valuable comments and suggestions.

Introduction, paragraph 2, what is the basis of the claim in line 1? Is there a missing citation?

The reviewer correctly noticed that a reference was missing. We have added the reference ([3]) supporting the statement that the rates of ASCVD are strikingly high in India as compared to western countries. This paragraph was written to highlight the burden of ASCVD in India and to point out that T2DM, in particular, contributes to this burden (last sentence of the paragraph) 

Research design and methods, statistical analysis, the authors mention 10 year, whereas the study was conducted for patients diagnosed between 2017-2020. Please clarify.

We understand that the sentence has caused confusion. We removed the part “10 years” and added an extra sentence; 

“The QRISK3 algorithm calculates a person's risk of developing a heart attack or stroke over the next 10 years”

What was the selection criteria for ‘newly diagnosed’? Were these patients diabetic for a long time and just discovered later? 

In line 1 under materials and methods we added (“i.e. first time”) diagnosed. To explain what is meant by this we also added the following sentence in the Discussion section under strength and limitations: 

“Newly diagnosed T2DM was defined as the first time the patient was diagnosed with T2DM. However, some patients might have had T2DM for a long time already that was only recently discovered, whereas other patients were diagnosed relatively earlier. This might have caused some heterogeneity within the present patient population”

What were the Hb1Ac levels for these patients?

We have added the average HBA1c values for both males and females to table 2 and we added the following sentence to the results 

“…., an average HBA1c value of 8.1%”

India is a very diverse country in terms of ethnicity. What were the ethnicities of the patients? Does that play a role in assessing the risk factors?

The reviewer has a valid point. Unfortunately, we did not verify this. However, it is known that there are regional differences in CV risk among Indians. As we consider both aspects relevant to consider when verifying CV risk, we have added the following sentences to “strength and limitations” in the discussion section: 

“India is a diverse country in terms of ethnicity, and it cannot be excluded that this might have some effect on cardiovascular risk. In addition, a recently published results of more than 100,000 Indians screened for CVD risk at airport locations showed that there were differences in BMI, blood glucose, and blood pressure values between some states of India [28]. Although, both ethnicity and region might be relevant and worth considering when verifying CVD risk among Indians, this was not done in the present study.”

Page 15, last line mentions male twice instead of male and female.

We thank the reviewer for this comment and have changed this into males and females. 

Some subheadings are italicized while others are not. Please correct it.

We have corrected the subheadings to Italic. 

Reviewer #2: Ref: PONE-D-21-35344

In the present article entitled “Cardiovascular risk in newly diagnosed type 2 diabetes patients in India” Unnikrishnan et al. have looked into the extent of cardiovascular disease (CVD) risk of newly diagnosed T2DM patients in India. The study design is simple and the results are well supported by the data.

The multicentric nature of the study makes the sample representative of the diverse population in a vast country like India. The limitations of the study have been laid out well.

Here are a few suggestions to make the manuscript more robust prior to publication:

We kindly thank the reviewer for the valuable comments and suggestions. 

1. Details of Funders of the study have not been included in the financial disclosures: 'Enter a financial disclosure statement that describes the sources of funding for the work included in this submission.'

We added a financial disclosure statement on P27 of the manuscript

2. Detailed ethics statement should be made available prior to final publishing since the study involves health data from multiple participants.

We have added both the Ethical approval from the Astha institute (S1 File) and Chellaram Diabetes Institute (where the principal investigator is affiliated to [S2 File) as a supplement and referred to the this under Material and Methods

3. Kindly inform the editorial team how we can help with regards to making the underlying the results presented in the study available : '... but some help from the journal may be needed.'

We more than willing to share the (anonymized) data based on reasonable request. I kindly propose the following: If there is e.g. a research group who would like to verify the data they can contact the corresponding author from whom the email address is provided in the paper. Upon which the corresponding author can send the data by email. We have added the following statement to the acknowledgements: 

“The data will be made available upon reasonable request. For this the corresponding author can be contact (willem.verberk@microlife.ch)”. 

Please note that this is just a proposal if the editor prefers another approach, we are happy to follow this. 

4. Kindly submit supplementary data regarding patient details, blood tests done/other parameters calculated.

We have added study protocol and CRF (S4 File) in the supplement material in addition, we have added the patient details at a separate file and the certificate of the Lab (S5 File) to the supplementary materials. 

5. The names of the participating doctors/ scientists may be included in a supplementary document

We have followed the reviewer’s instruction and added this to the supplement (S1 File) and referred to this in the text. 

Reviewer #3: In this manuscript authors are showing the cardiovascular risk assessment in newly diagnosed T2DM patients in India. Sample size and the guidelines to assess the risk is appropriate for this cross-sectional study. except minor edits, manuscript looks good.

We kindly thank the reviewer for his/her valuable time for reviewing this manuscript

---

## [Decision Letter · Decision Letter 1]

24 Jan 2022

Cardiovascular risk in newly diagnosed type 2 diabetes patients in India.

PONE-D-21-35344R1

Dear Dr. Verberk,

We’re pleased to inform you that your manuscript has been judged scientifically suitable for publication and will be formally accepted for publication once it meets all outstanding technical requirements.

Kind regards,

Kanhaiya Singh, Ph.D

Academic Editor

PLOS ONE

Additional Editor Comments (optional):

Reviewers' comments:

Reviewer's Responses to Questions

**Comments to the Author**

1. If the authors have adequately addressed your comments raised in a previous round of review and you feel that this manuscript is now acceptable for publication, you may indicate that here to bypass the “Comments to the Author” section, enter your conflict of interest statement in the “Confidential to Editor” section, and submit your "Accept" recommendation.

Reviewer #1: All comments have been addressed

Reviewer #2: All comments have been addressed

2. Is the manuscript technically sound, and do the data support the conclusions?

Reviewer #1: Yes

Reviewer #2: Yes

3. Has the statistical analysis been performed appropriately and rigorously? 

Reviewer #1: Yes

Reviewer #2: I Don't Know

4. Have the authors made all data underlying the findings in their manuscript fully available?

Reviewer #1: Yes

Reviewer #2: Yes

5. Is the manuscript presented in an intelligible fashion and written in standard English?

Reviewer #1: Yes

Reviewer #2: Yes

6. Review Comments to the Author

Reviewer #1: All the reviewers comments have been addressed.

Reviewer #2: (No Response)

7. PLOS authors have the option to publish the peer review history of their article (what does this mean?). If published, this will include your full peer review and any attached files.

Reviewer #1: No

Reviewer #2: **Yes: **Tejas Nimba Nikumbh

---

## [Editor Report · Acceptance letter]

15 Feb 2022

PONE-D-21-35344R1 

Cardiovascular risk in newly diagnosed type 2 diabetes patients in India. 

Dear Dr. Verberk:

I'm pleased to inform you that your manuscript has been deemed suitable for publication in PLOS ONE. Congratulations! Your manuscript is now with our production department. 

Kind regards, 

on behalf of

Dr. Kanhaiya Singh 

Academic Editor

PLOS ONE